1

2

# Simulations of water, heat, and solute transport in partially frozen soils

- Mousong Wu<sup>1,2</sup>, Per-Erik Jansson<sup>2</sup>, Xiao Tan<sup>1</sup>, Jiesheng Huang<sup>1</sup>, Jingwei Wu<sup>1</sup>
- 1.State Key Laboratory of Water Resources and Hydropower Engineering Science, Wuhan
   University, 430072 Wuhan, Hubei, China

2.Department of Sustainable Development, Environmental Science and Engineering, KTH
 Royal Institute of Technology, 10044 Stockholm, Sweden

# 7 Abstract

8 Experiments for soil freezing/thawing were conducted in two seasonally frozen agricultural fields in northern China during 2011/2012 and 2012/2013 wintertime, respectively. Mass 9 balance was checked based on measured data at various depths. Simulation work was 10 conducted by combining CoupModel with Monte-Carlo sampling method to achieve 11 parameter sets with equally good performance. Uncertainties existed in both measurements 12 and model due to complexity in freezing/thawing processes as well as in surface energy 13 partitioning. Parameters related to surface radiation and soil frost were strongly constrained 14 with datasets available in two sites combining multi-criterion on outputs. Simulated soil heat 15 processes were better described than soil water processes given the data obtained for 16 calibration. Model performance was improved with consideration of solute effects on 17 freezing point depression. More detailed solute transport processes in CoupModel needed to 18 be improved by taking more processes such as diffusion and expulsion into consideration 19 based on more precise experimental results, to reduce uncertainty in model. Generally, 20 combination of measurement with process-based model and Monte-Carlo sampling method 21 provided an approach for understanding of solute transport as well as its influences on soil 22 freezing/thawing in cold arid agricultural regions. Incorporating more detailed descriptions of 23 processes for frozen soil in the model can be justified if uncertainties in measurements can be 24 reduced by introducing of high-precision novel technologies. 25

26

Keywords: Frozen soil; Solute; Uncertainty; Freezing point; Salinization

Correspondence author. Email: jingwei.wu@whu.edu.cn

### 28 1. Introduction

Soil freezing and thawing processes has long been recognized for its importance in not only engineering applications (e.g., construction of roads and pipelines) (Hansson et al., 30 2004; Wettlaufer and Worster, 2006; Jones, 1981), but also environmental issues (e.g., soil 31 erosion, flooding, and pollutants migration) (Seyfried and Murdock, 1997; Andersland et al., 32 1996; Baker and Spaans, 1997; McCauley et al., 2002). The investigation of soil freezing and 33 thawing could result in a better understanding of water and solute distribution in soil (Baker 34 35 and Osterkamp, 1989), frost heaving (Wettlaufer and Worster, 2006), waste disposal (McCauley et al., 2002), climate change in cold regions (Lopez et al., 2007). 36

Many experimental observations have been conducted since 1930s to study the 37 freezing/thawing phenomenon in soil (Beskow, 1947; Edlefsen and Anderson, 1943; Spaans 38 and Baker, 1996; Miller, 1980). For example, Beskow (1947) observed the accumulation of 39 water towards freezing front and the influences of water, soil and solutes on freezing point 40 depression, as well as the similarity between freezing and drying. Edlefsen and Anderson 41 (1943) then formulated the relationship between soil temperature and freezing soil water 42 43 potential by generalized Clausius-Clapeyron equation. Koopmans and Miller (1966) then tested the similarity between soil freezing curves and water retention curves. Burt and 44 Williams (1976) measured hydraulic conductivity of freezing soil in laboratory, and this work 45 was then put forward by others (Nakano et al., 1982; Horiguchi and Miller, 1983; Black and 46 Hardenberg, 1991). At the same time, formulation of the soil freezing/thawing processes has 47 been raised up. e.g., the use of Clausius-Clapeyron equation for representation of soil 48 freezing equilibrium (Kay and Groenevelt, 1974; Groenevelt and Kay, 1974), the power 49 relationship between liquid water content and soil temperature (Anderson and Tice, 1973), 50 the capillary bundle model for soil freezing characteristics (Watanabe and Flury, 2008; 51 Lebeau and Konrad, 2012), and the influences of solute on soil freezing/thawing (Bing and 52 Ma, 2011; Azmatch et al., 2012; Wu et al., 2015). 53

Laboratory and field experiments on soil freezing/thawing processes have received more 54 55 attention. Watanabe et al. (2013) conducted laboratory experiments to describe the influence 56 of soil freezing on infiltration. Zhou et al. (2014) measured the water content and ice content in a freezing soil column with gamma ray attenuation and TDR method. Also, field 57 experiments were conducted to study plot-scale or regional water, heat and solute transport 58 during freezing/thawing from different aspects. Radke and Berry (1998) analyzed the 59 influences of soil water, bulk density as well as microbial activities on soil water and solute 60 transport in the field soil column experiments. Stahli et al. (2004) characterized the 61

preferential flow in frozen soil with dying method and proposed a two-domain model for flow in frozen soil. Iwata et al. (2008, 2010a, 2010b) studied the infiltration of snow melt 63 under various controls as well the influences on water, heat dynamics in frozen soil, in the 64 agricultural field in Japan. Parkin et al. (2013) studied the effects of tillage on 65 freezing/thawing of agricultural field. Zhao et al. (2013) conducted field experiment to 66 analyze influence of snowmelt infiltration on hydrological processes in winter in Inner 67 Mongolia, China. Wu et al. (2016) studied the evaporation from seasonally frozen soil under 68 various salt and groundwater conditions using field frost tube experiments, in Inner 69 Mongolia, China. All these experiments demonstrated that water, heat and solute in frozen 70 71 soil could be influenced by both soil conditions and boundary conditions, but due to a lot of uncertainties in experimental treatments as well in measurements, experimental studies on 72 73 frozen soil could result in some uncertainties in knowledge of soil freezing/thawing.

Along with the experimental studies, numerical models have been put forward by many. The coupled water and heat transport model by Harlan (1973) considered the coupled 75 relationship between water and heat in frozen soil. Then, Jame and Norum (1980) set up a 76 77 finite element numerical model based on Harlan's model, and tested it with laboratory experimental results. This coupled model was then improved and tested by a lot of 78 researchers with datasets from both laboratory and field (Mu and Ladanyi, 1987; Li et al., 79 2000; Li et al., 1998). All these models did not consider the transport of solute in frozen soil 80 and neglected the influence of solutes on soil freezing. Flerchinger and Saxton (1989) 81 proposed a simultaneous heat and water model for simulating water, heat and solute transport 82 in frozen soil with snow and residue covering. Then this model was tested in many cold 83 regions (Li et al., 2012; Li et al., 2013) and showed high flexibility in application under 84 various conditions. Jansson and Karlberg (2004) developed a coupled process-based model 85 based on the SOIL model to simulate water, heat as well solute transport in frozen soil. This 86 model was verified in forests (Gustafsson et al., 2004; Wu and Jansson, 2013), agricultural 87 field (Wu et al., 2011), permafrost (Zhang et al., 2012; Scherler et al., 2013) and other 88 89 ecosystems (Khoshkhoo et al., 2015). Hansson et al. (2004) also added a freezing module to 90 one-dimensional water, heat and solute transport model HYDRUS, and tested the sensitivity of model using experimental results. Meanwhile, a lot of other models have taken soil 91 freezing/thawing into consideration when applied to wintertime (e.g., SWAP, DRAINMOD, 92 SWAT, HBV, VIC etc.). Numerical models have become a popular tool for understanding 93 water, heat as well as solute transport in winter with complex boundary conditions and phase 94 95 change.

96 However, there are large uncertainties in both experiments and models for soil freezing 97 and thawing due to the complexity of phase change and coupled processes. For example, the measurement of liquid water content in frozen soil, the sampling of frozen soil, the 98 measurements of hydraulic and thermal properties for frozen soil could be difficult due to 99 limitations in technologies and in considering all effects on soil freezing/thawing (e.g., water, 100 101 heat, solutes, soil textures as well as boundary conditions). Meanwhile, the setup of model always neglected some minor influences by taking the major one into consideration, e.g., the 102 assumption of thermo-equilibrium of soil freezing, the neglecting of solute dispersion and 103 expulsion in frozen soil, or even the neglecting of solute effects on freezing point depression, 104 etc. All these would pose uncertainties to the study on soil freezing/thawing in natural 105 conditions. To reduce uncertainties in both experiments and modeling, uncertainty analysis 106 method is always used by combining experimental data with numerical model to calibrate the 107 model for better representing reality. The generalized likelihood uncertainty estimation 108 (GLUE) technique (Beven and Binley, 1992) is the commonly used method for uncertainty 109 analysis in environmental modeling. 110

111 Instead of searching for an optimal parameter set, the GLUE method generates ensembles of parameter sets that show equally good performance in simulations (Candela et 112 al., 2005), as called 'equifinality' by Beven (2006). This method has been widely used in 113 hydrologic simulations (Freer et al., 1996; Beven and Freer, 2001; Liu et al., 2009; Li et al., 114 2010; Song et al., 2015; Sun et al., 2016) and climate change projections (Cameron et al., 115 2000; Wilby, 2005; Choi and Beven, 2007; Bastola et al., 2011; Lin et al., 2015). There are 116 only a few modeling work with agricultural water resources (Brazier et al., 2000; Wang et al., 117 2006; He et al., 2010; Wu et al., 2011; DeJonge et al., 2012; Chisanga et al., 2015). 118

In models for soil water, heat and solute transport (e.g., CoupModel, HYDRUS, 119 SHAW), there are many parameters related to different coupled transport processes, also 120 including non-linear responses, especially when considering soil freezing/thawing and solute 121 transport. The parameters in these models are possible to measure with independent methods 122 123 but those are big challenges because of high variability in the environments and many 124 temporal and spatial scale related dependencies. Also, model structures are always simplified for description of some processes. Thus, for investigation of coupled processes in seasonally 125 frozen agricultural field, it is important to use the GLUE method combining process-based 126 model to unveil the freezing/thawing phenomenon. 127

In this study, we conducted field experiments on water, heat and solute transport in two sites in northern part of China. They are special both for the climates and the soils in cold

regions in China. The coupled transport of water, heat and solute during wintertime is
common for these two sites. Also, due to limited tools in measurements, only the common
variables were observed using common methods during experiments.

The main objective was to search for constrain with the current available data and models by considering also explicit salinity impacts on freezing in the model. With the collected data from field, the model was combined with Monte-Carlo sampling method to 1) investigate how well simulated water and solute dynamics corresponded to measured data; 2) identify variability in modeling of water, heat and solute using ensemble simulations; and 3) discuss the influences of solute on soil freezing/thawing based modified freezing point depression function.

140

### 141 **2. Material and methods**

2.1 Study sites

Studies were conducted at two experimental fields in north China, during 2011/2012, 143 and 2012/2013 wintertime, respectively. One site was located in Qianguo Irrigation District, 144 Songyuan, Jilin (Site NE) (Fig. 1). Annual precipitation in Site I was 451 mm and mean 145 monthly air temperature was 5.1 °C. The study site is typical for its soil texture of clay, which 146 has a high bulk density, low porosity, and low hydraulic conductivity (Table 1). The water 147 table in this area fluctuated between 1.5 and 2.0 m. Maximum frost depth in Site I was 1.2 m. 148 In Site I, six plots  $(2 \times 2 \text{ m}^2)$ , denoted as P1 to P6) were selected in an agricultural field, which 149 was cultivated with rice from May to October. On 2011/10/09, 20 mm NaBr solution 150 containing 6.5 g L<sup>-1</sup> Br<sup>-</sup> was applied to six plots to form the initial profile for Br<sup>-</sup>. Before 151 spraying of the solution, stubbles were removed from the plots and surface was ploughed to 152 depth of 20 cm. Field experiment in Site I was conducted during 2011/2012 wintertime. 153

The other site was located in Hetao Irrigation District, Inner Mongolia, China (Site IM) 154 (Fig. 1). Annual precipitation in this site was 140 mm. Annual mean air temperature 6.4 °C. 155 Soil texture in this site is characterized as silt loam, with porosity of 0.42~0.46 and saturated 156 hydraulic conductivity of  $3.84 \times 10^{-5}$  m s<sup>-1</sup>. Water table was kept between 1.5 and 3 m for the 157 winter time, and three irrigation events occur every year in May, July, and November. Soil 158 salt content (mainly NaCl) was 0.1% g g<sup>-1</sup> for the study field, and irrigation water with 159 electrical conductivity of 0.5 mS cm<sup>-1</sup>. Before autumn irrigation, five plots ( $2 \times 2$  m<sup>2</sup>, denoted 160 as D1-D5) were selected at different parts of the agricultural field, and ploughed to 20 cm 161 depth. Field experiment in this site was conducted during 2012/2013 wintertime. 162

# 165 2.2 Experimental design

TDR probes with 15-cm long and three-rod (CS605) were installed in Site NE to detect 166 liquid water content. A datalogger (TDR 100; Campbell Scientific Inc.) was connected to the 167 probes and recorded daily water data. TDR probes were inserted horizontally into the soil pit 168 (10 m apart from the plots) from 5 cm to 100 cm with 10 cm interval. TDR probes were 169 calibrated in laboratory with unfrozen soil, and the precision was maintained within  $R^2$  of 170 0.97. PT100 temperature sensors were installed at the same depth as TDR probes, and the 171 daily temperature data was also collected. During soil freezing/thawing, sampling was 172 conducted at 7 dates (2011/10/09, 2011/11/09, 2011/11/25, 2011/12/20, 2012/02/15, 173 2012/04/10, 2012/04/20), and soil samples from 5 to 100 cm with 10 cm interval were 174 collected for determining total water content and Br content. An electric drill (5 cm in 175 diameter, 10 cm in length) was used for sampling soil from depth to depth. Total water 176 content was determined by oven-dry method. Br content was determined by diluting 50 g 177 wet soil into 250 mL deionized water, and measuring the electrical potential (mV) using an 178 electrical potential meter (MP523-06). Then the electrical potential was converted into Br 179 concentration by a pre-calibrated relationship between Br concentration and electrical 180 potential ( $R^2$ =0.99). Total water content and Cl<sup>-</sup> content in Site IM were sampled at 14 dates 181 (with around 25-d interval) from October 2012 to April 2013 (2012/10/16, 2012/10/27, 182 2012/11/10, 2012/12/04, 2012/12/15, 2012/12/26, 2013/01/05, 2013/01/14, 2013/01/25, 183 2013/03/05, 2013/03/14, 2013/03/25, 2013/04/07, 2013/04/18). The sampling and 184 measurement methods for total water content and Cl<sup>-</sup> content were the same with those in Site 185 NE. Hourly soil temperatures at 5, 15, 25 and 35 cm depth were recorded by the PT100 186 temperature sensors from the micro-meteorological station in the field. Groundwater table 187 depth was measured for every 5 d, and the frequency was increased to every 1 d during the 188 autumn irrigation period (2012/11/4 to 2012/11/15). Meteorological data for two sites, e.g., 189 air temperature, humidity, radiation, wind speed, and precipitation, were obtained from the 190 191 nearest meteorological station with hourly-resolution.

# 193 2.3 CoupModel theory

In seasonally frozen soil, the transport of water, heat, and solute in soil profile is coupled with lower boundary (groundwater) and upper boundary (atmosphere) (**Fig. 2**). Water transport processes in CoupModel could be described by combining Darcy's law with mass conservation law:

$$\frac{\partial \theta}{\partial t} = \frac{\partial}{\partial z} \left[ k_w \left( \frac{\partial \psi}{\partial z} - 1 \right) \right] + \frac{\partial}{\partial z} \left( D_v \frac{\partial C_v}{\partial z} \right) - \frac{\partial q_{bypass}}{\partial z}$$
(1)

where  $\theta$  is water content (m<sup>3</sup> m<sup>-3</sup>);  $k_{\psi}$  is hydraulic conductivity (m s<sup>-1</sup>);  $\psi$  is matric potential (m);  $D_{\psi}$  is vapor diffusion coefficient (m<sup>2</sup> s<sup>-1</sup>);  $C_{\psi}$  is vapor density (kg m<sup>-3</sup>);  $q_{bypass}$ is the bypass flow in macro pores (m s<sup>-1</sup>); z is depth to soil surface (positive downward) (m); and t is time (s).

Unsaturated hydraulic conductivity,  $k_w$ , is calculated by Mualem equation (Mualem, 1976) combined with Brooks-Corey water retention curve (Brooks and Corey, 1964). Hydraulic conductivity in freezing/thawing soil is modified by dividing water flow domain into high flow and low flow domains (Stähli *et al.*, 1996), and adjusted by using impedance factors, respectively.

Heat flow in soil is described by the heat transport equation, considering conduction, convection and latent heat flow:

$$\frac{\partial(CT)}{\partial t} - L_f \rho_i \frac{\partial \theta_i}{\partial t} = \frac{\partial}{\partial z} \left( k_h \frac{\partial T}{\partial z} \right) - C_w T \frac{\partial q_w}{\partial z} - L_v \frac{\partial q_v}{\partial z}$$
(2)

where *C* is soil (containing solid, water, and ice) heat capacity (J m<sup>-3</sup> °C<sup>-1</sup>); *T* is temperature (°C);  $L_f$  is latent heat of freezing (J kg<sup>-1</sup>);  $\rho_i$  is density of ice (kg m<sup>-3</sup>);  $\theta_i$  is ice content (m<sup>3</sup> m<sup>-3</sup>);  $k_h$  is thermal conductivity soil (W m<sup>-1</sup> °C<sup>-1</sup>);  $q_w$  is water flux (m s<sup>-1</sup>);  $L_v$  is latent heat of vaporization (J kg<sup>-1</sup>); and  $q_v$  is vapor flux (m s<sup>-1</sup>).

Thermal conductivity for frozen/unfrozen soil is calculated by the Balland and Arp (2005) method considering the influences of soil components on heat flow.

Solute in CoupModel is considered to transport with water, neglecting diffusion. Solute 218 transport is converted into Cl<sup>-</sup> transport in soil:

$$q_{Cl} = c_{Cl} q_w \tag{3}$$

where  $c_{cl}$  is concentration of Cl<sup>-</sup> (kg m<sup>-3</sup>); and  $q_w$  is water flux (m s<sup>-1</sup>).

Upper boundary for model is atmosphere and snow layer is also taken into consideration. Lower boundary is saturated soil layer and drainage is calculated by the combination of empirical drainage equation with Hooghoudt drainage equation The detailed descriptions of water, heat, and solute transport processes as well as model boundaries could be found in the CoupModel manual (Jansson and Karlberg, 2004). Equations used for this simulation work were detailed listed in **Table S2** in Appendix.

219

2.4 Modification of freezing point depression functions

In CoupModel, the freezing-point depression (**Fig. 3**(a)) is described as below:

$$r = \left(1 - \frac{E}{E_f}\right)^{d_2 \lambda + d_3} \min\left(1, \frac{E_f - E}{E_f + L_f w_{ice}}\right)$$
(4)

where  $d_2$ ,  $d_3$  are empirical constants,  $\lambda$  is the pore size distribution index.

In saline frozen soil, ice formation does not start at 0 °C, but at freezing point of  $T_0$ .  $T_0$ is a parameter related to soil type, solute type and solute content. In this approach,  $T_0$  needs to be defined as a parameter in CoupModel, and this parameter could be determined based on experiments or by calibration with soil temperature data for frozen soil. In the present version of CoupModel,  $T_0$  is actually taken as 0 °C, as shown in **Fig. 3**(a). In this development of the model,  $T_0$  is introduced with values below 0 (from -3 to 0 °C), the freezing point depression will be like in **Fig. 3**(b).

When calibrating the model in terms of freezing/thawing, parameter  $T_0$  needs to be determined with respect to salt influence on soil freezing/thawing.

As the influence of salt on freezing point is mainly dependent on the osmotic potential of soil solution, a third relationship between freezing point and osmotic potential is built. According to Banin and Anderson (1974), the relationship between freezing point and salt solution could be written as below:

 $T_0 = -1.86c \frac{N_m}{Z} \tag{5}$ 

where *c* is salt concentration (mol  $L^{-1}$ );  $N_m$  is number of ions to which the salt molecule dissociates; *Z* is valency of salt.

In CoupModel, osmotic potential ( $\pi$  (cm)) is a function of salt concentration 249  $\pi = 10^{-3} \times R(T + 273.15) \cdot c$  (6)

where *R* is gas constant (8.31 J mol<sup>-1</sup> K<sup>-1</sup>); *T* is temperature (°C); *c* is salt concentration (mol L<sup>-1</sup>).

Substituting Equation (6) to Equation (5) will obtain:
$$T_0 = -1.86 \times 10^3 \cdot \frac{N_m}{Z} \cdot \frac{\pi}{R(T+273.15)}$$
(7)

For simplicity, Equation (7) could be expressed as below with a linear relationship between freezing point and osmotic potential:

$$T_0 = -10^{-4+sc} \times \frac{\pi}{1.221} \tag{8}$$

where  $T_0$  is the freezing point (°C);  $\pi$  is osmotic potential (cm); *sc* is a scale factor for considering the influences of solute types on the relationship (range from -2 to 2); -4 is a constant for converting osmotic potential unit from cm to MPa.

## 261 2.5 Uncertainty analysis approach

1) Model performance indicator selection

The likelihood is necessary for a formal objective selection of accepted simulations 263 based on the uncertainty of also the measurement method. However, this requires information 264 about various errors in the measurement approach, which is tricky to obtain without making 265 very extensive investigations. There are many different kinds of substitutes for likelihood 266 functions (Beven and Freer, 2001) when evaluating model performance, and they are widely 267 used in hydrology (e.g., Li et al., 2010; Besalatpour et al., 2012; Pathak et al., 2012). Here, 268 the Nash-Sutcliff index was selected as a performance indicator to be used to reject non-269 behavioral simulations. This function is calculated as follows: 270

271 
$$L(\theta | Y) = \text{NSE } R^2 = 1 - \frac{\sum_{i=1}^{N} (Y_i - \hat{Y}_i)^2}{\sum_{i=1}^{N} (Y_i - \overline{Y})^2}$$
(9)

where  $L(\theta | Y)$  is the indicator for each model run with parameter set  $\theta$ , N is the total number of measurements,  $Y_i$  is the measured value for the *i*th measurement and  $\hat{Y}_i$  is the corresponding output of the model,  $\overline{Y}$  is the average of the measured data. If the model predicts the measurements perfectly, we have  $Y_i = \hat{Y}_i$ , implying NSE  $R^2 = 1$ . If  $\hat{Y}_i = \overline{Y}$  for all *i*, then NSE  $R^2 = 0$  has the same goodness of fit as using the average of the measured data for every situation.

278

#### 279 2) Model calibration with Monte-Carlo sampling method

In this work, we calibrated the model in two study sites using data from one winter period within the GLUE framework, respectively. Since this study is to discuss the model performance and parameter uncertainty for a process-based model in simulation of water, heat and solute transport in frozen soils, we would focus more on the calibration of the model instead of validation.

In a physically based model assuming possible variability between soil layers, around 285 100 parameters are used for simulation. However, for most parameters, they are not 286 necessarily to be calibrated according to the interest of the modeling work. Thus, in this 287 study, a set of parameters that show high sensitivities were selected for calibration, as shown 288 in Table S1. These parameters were selected on the basis of one-parameter-at-a-time 289 sensitivity analysis, which was conducted before setting up the model. For the other 290 parameters, they were set as fixed values based on experimental results or references to 291 292 previous research (Wu et al., 2011a; Gustafsson et al., 2001; Metzger et al., 2015). Then, a uniform prior distribution was assigned to each parameter and 70,000 random parameter sets 293

were created by Monte-Carlo sampling method.

The simulated variables for two sites (temperature, liquid water content at 5, 15, 25, and 295 35 cm depth for Site NE; temperature, total water content at 5, 15, 25, and 35 cm depth, and 296 groundwater level for Site IM) were compared with mean of measured from multi-plot, and 297 NSE  $R^2$  was used to constrain model performance to achieve the accepted simulations. At 298 Site NE, NSE  $R^2 > 0.7$  for soil temperature at multi-depth and NSE  $R^2 > -1$  for liquid water 299 content at multi-depth were chosen to constrain the simulations. At Site IM, NSE  $R^2 > 0.8$ , 300 NSE  $R^2 > 0.5$ , and NSE  $R^2 > 0.5$  were used for constraining temperature, total water content 301 and groundwater table depth, respectively. These criteria were chosen based on the 302 cumulative distribution of the model performance, by reducing the number of accepted 303 simulations to around 200, and keeping the model performance (NSE  $R^2$ ) of accepted 304 simulations higher than 80% of all the simulations. Finally, 204 and 222 accepted simulations 305 were obtained for Site NE and Site IM, respectively. 306

307

# 308 3. Results and discussion

#### 309 *3.1 Water and solute balance analysis*

Changes of water and solute storage for different soil horizons showed typical patterns 310 (Fig. 4). Seven sampling dates at Site NE divided the whole experimental duration into six 311 periods. While at Site IM, 13 sub-periods were obtained by 14 sampling dates during the 312 experiment. At Site NE, water storage tended to increase from Period 1 to 5 (Fig. 4(a)), for 0-313 10, 0-40, and 0-100 soil zones, except for Periods 2 and 3, when water storage tended to 314 decrease for some zones. For Period 2, water storage decrease in 0-10 cm zone was mainly 315 due to evaporation loss because solute storage (Fig. 4(b)) in 0-10 cm zone of this period 316 increased. Besides, solute storage change in three zones during Period 2 were similar, which 317 meant that water and solute change in soil layer lower than 10 cm depth did not influence 318 water and solute storage change in 0-10 cm zone. 319

Similarly, in Period 3, water loss in 0-10 cm soil layer was also due to evaporation. In Periods 4 and 5, water storage increased largely in soil profile, because of upward movement of water under temperature and potential gradients during soil freezing. In Period 6, water storage in 0-100 cm depth generally decreased, because in this period, soil was thawing, evaporation, runoff would cause large amounts of water loss from soil profile. Solute storage in Periods 5 and 6 showed less change or slight decrease due to loss of water from soil profile.

Water storage at Site NE (Fig. 4(c)) generally increased for most periods, with Period 2