# Peer review of "Simulations of water, heat, and solute transport in partially frozen soils"

_Hydrology and Earth System Sciences, 2016_

## Referee Comment (RC1) · Anonymous Referee #1 · 5 Dec 2016

General comments

Wu et al. conduct a GLUE-type sensitivity analyses using a model of coupled heat, water, and solute transport. The study has some merit and potentially fills a gap. I'll not deny that there are very few sensitivity studies of these phenomena; however, this paper is poorly presented both in terms of the technical information as a well as the overall story.

Major comments

1. The English in this paper is not, in my opinion, even suitable for the first submission, let alone consideration for publication. It must be rewritten by a professional English editing company. I originally began to do this for the authors, but got exhausted by about L140.

[Figure]

2. Other places are grammatically correct, but incredibly vague. The authors must carefully reread through this study and make sure their sentences convey meaning. For example, 'Laboratory and field experiments on soil freezing/thawing have received more attention' (L54); what does this mean? More than what? Why is this needed? Similarly go to L36: "climate change in cold regions"? What about climate change? How does it relate to solute? Explain. This doesn't fit the sentence. There are many examples of such vague statements with little to no information (e.g., L286-287). I don't see this as an English problem, but rather as a contribution that needs to be carefully rewritten in general.

3. The introduction does not build a convincing story of why this contribution is needed. First of all it is too long (9 paragraphs). As an example of extraneous text, in Lines 37 to 95 the authors list a number of soil freeze-thaw field and modeling studies and where those studies were conducted. However, they really don't emphasize the contribution or key input of over half of these studies. These are described in far more detail in the review paper by Kurylyk and Watanabe (2013), and listing fewer of these and referring to this synthesis would be a more effective use of space. More importantly, the main objective (e.g. 'search for constrain' L133), makes no sense, and thus it is very hard to get excited about the rest of the paper.

4. The title does not have 'uncertainty' in it. In fact, the title could probably equally apply to about 12 of the other papers cited in this study. My point is that it should be rewritten somehow to reflect the distinct aspects of this study.

5. L168, Why did the authors record TDR data at daily intervals? This seems like a low resolution given the frequency of the other data. Depending on the depth and spatiotemporal resolution, it can be very hard to calibrate or assess a model using only daily moisture data.

6. There are issues with the only two equations I looked at very closely on a term by term basis. Equation 1: The terms in this equation do not have consistent units.
Therefore something is clearly wrong. I think the second term on the right (vapor diffusion) has incorrect units and the vapor density should be expressed as a vol/vol. Equation 2: This equation is expressed incorrectly for the divergence of convection (second term on right hand side). The temperature has to be inside the derivative. It is changing with space. Also, if you leave it outside the derivative, you have the issue that it totally depends on what temperature scale you are using (Celsius vs. Kelvin) in terms of the magnitude of that term. All other terms are independent of the temperature scale, because it is only the change in temperature that matters (i.e. the other T terms are inside derivatives). Two errors in the two equation carefully considered does not give one confidence in the rest of the paper. I strongly recommend that the authors go through the equation appendix very carefully.

7. The figures are poorly done in general. Is Figure 2 taken from the Coup manual, at least in revised form? If so, that should be stated. Figure 3 is unclear. What is energy? Is this sensible and latent heat? What do the different colors represent? Figure 4 is confusing. Why is the cross-hatching so similar? Are these cumulative? For example, for time period 2, the 0-100 section goes from 0 to -18 (I think), but the next one goes from -18 to about -38. What does this represent? Explain in the caption!

8. How does this paper differ from Wu et al. (2016) by the same authors. It is an uncertainty study using a similar sort of model it would seem. I'm not saying there are no differences, but it seems, at least surficially, to be similar. Wu, M., P.-E. Jansson, X. Tan, J. Wu, and J. Huang. 2016. Constraining parameter uncertainty in simulations of water and heat dynamics in seasonally frozen soil using limited observed data. Water 8(2):64, doi:10.3390/w8020064

Oddly enough this is never cited. Also, the authors have a lazy reference list. For example, Wu et al. has been published in Cold Reg. Sci. Technol. for over a year now, but the authors list it as accepted. The cited Wu et al. 2016 study has no journal information, so it is not easy to look up. Again, those were the only two reference items that I looked at.

These are all very short, general comments, but a more careful, rigorous review is really difficult given the present state of the manuscript.

In summary, this is certainly not publishable in its current form, but I think it does have potential for publication in some journal some day given the content.

―――――――――――――

---

## Referee Comment (RC2) · Anonymous Referee #2 · 4 Jan 2017

General comments

The manuscript send by Wu et all provides a study of simulating water and solute infiltration in partially frozen agricultural soil in northern China. Two winter periods were simulated. They used 1D subsurface, surface and atmosphere model (CoupModel) that is capable of simulating heat transfer, and soil water distribution in frozen and partially frozen soil. CoupModel is highly tested and quite well recognize model in cold snow dominated regions, and can be the most sophisticated model to assess water infiltration in partially frozen soils in cold regions.

The author uses novel technique in calibrating the highly parameterizes model. Even though Monte Carlo method is well used and has been (and should be) a standard process in every simulation exercise this paper provides some new information of sensitivity of parameters affecting simulated soil water content and temperature (heat transfer) in frozen soils, new insight is also given to account solute in the model calibration, although it seems that CoupModel cannot provide a very good solute transport solvers (as author also clearly recognize this) and there can be better tools available for that. This study also reveals that where the pitfalls are if solute transport is taken into account, and show that the results can be improved if solute transport is taken into account.

There is some moderate/major issues that the authors need to take into account prior to publication.

Abstract

Lines 24-25, the author mention "novel techniques" but this study do not explore these techniques in depth, this needs to be removed from the abstract because is not really the scope of the study.

Introduction

In recent literature there has been debate of using impedance factor, that CoupModel uses, in hydraulic conductivity function to account the blocking effect of ice (see Painter et al. 2016; Kuryluk and Watanabe 2013). The author could shorty improve introduction and discuss about this matter, because it has been raised up in the recent literature.

REF

Kurylyk, B. L., and K. Watanabe (2013), The mathematical representation of freezing and thawing processes in variably-saturated, non-deformable soils, Adv. Water Resour., 60, 160–177

Painter, S. L., E. T. Coon, A. L. Atchley, M. Berndt, R. Garimella, J. D. Moulton, D. Svyatskiy, and C. J. Wilson (2016), Integrated surface/subsurface permafrost thermal hydrology: Model formulation and proof-of-concept simulations, Water Resour. Res., 52, 6062–6077, doi:10.1002/2015WR018427.
In lines 54-73, the author brings up a lot of literature about previous studies but do not mention the most important results of them. Maybe the most relevant studies with respect to this study should only bring up, and the most important results of them and how they are correlated or not with this study. The author mention very lightly (see line 71) the most common results: that soil condition and boundary conditions have an effect but this is way too general outcome of these previous studies. Line 93, ATS model should be mention in this list (see Painter et al 2016, WRR paper) because they have freezing/thawing included in distributed hydrological modeling.

Line 103, delete "the" before neglecting

Line 104, delete "the" before neglecting

Line 121 " , " delete empty space

Materials and Methods

Line 187-191, why the observation interval was not denser and not the same as observations from the meteorological station?

Line 193, delete "theory", because it is not only CoupModel theory but theory in general (flow etc..), maybe the author could invent a new headline.

Line 195, why groundwater level was chose as a lower boundary condition and not e.g. bedrock?

Line 196 "CoupModel could be. . ." what this "could" means? maybe "could" should be deleted

Line 199 and line 203, check the meaning of kw

Line 217, the CoupModel neglects diffusion, maybe the author could discuss about the effect of neglecting diffusion a little bit more in detail, the simulated results are not that good and maybe this is the major issue here.

[Figure]

Line 222-223, is the lower boundary "groundwater" level or "drainage", I think the choice of groundwater level, or drainage may affect the results. If drainage is used, then the drainage equation should be clearly showed. The author only mention that this equation appears in the table S2.

Line 232, T0 ?, please refer to equation

Line 284, what the author mean that you focus on calibration and not validation? What do you mean by validation?

Results and discussion

Line 341 How "temperature gradients" affect the soil water and solute transport, please explain.

Line 350, define NP

Line 365- 371, this is difficult to follow, if the reader is not aware of the equations, maybe the equation should be explained in the text, at least a proper references should be provide that reader immediately knows what equations should be looking at in the appendix.

Line 377-380, what this means in practice? Be more specific.

Lines 381-384, what this means in practice? Be more specific.

Line 388-394, the resolution (time interval) of observations seems to be very important, why not taking data hourly?

400-403, can this be also an issue of the model, see lines 205-207?

Line 436-437 unclear, please reword

Line 440-441 unclear, please reword

Line 458, "Simulated liquid water content. . ."? delete "in comparison with the measured"

[Figure]

Line 466-467, unclear, please reword. The equations should be clearly showed or referred.

Line 482 "validation", calibration instead?

Line 485 delete "work"

Line 485 "surface water"? do you mean "soil water"?

Line 484-487 unclear, please reword

Line 494, delete "work" after calibration

Line 496 change "assumption of" to "assumption that"

Line 504-505 unclear, please reword

Line 517 carefully. , delete "."

Line 517-518, be specific that what type of data should be collected.

Line 555 "surface water"? do you mean "soil water"?
* * *

---

## Author Comment (AC1) · 21 Feb 2017

REF1 comments General comments Wu et al. conduct a GLUE-type sensitivity analyses using a model of coupled heat, water, and solute transport. The study has some merit and potentially fills a gap. I will not deny that there are very few sensitivity studies of these phenomena; however, this paper is poorly presented both in terms of the technical information as well as the overall story.

Major comments 1. The English in this paper is not, in my opinion, even suitable for the first submission, let alone consideration for publication. It must be rewritten by a professional English editing company. I originally began to do this for the authors, but got exhausted by about L140.

Re: Thanks for the comments on English. We will take it seriously and ask for some

native speakers to help us in English expression. Meanwhile, we will send it to professional English editing company for further improvement.

2. Other places are grammatically correct, but incredibly vague. The authors must carefully reread through this study and make sure their sentences convey meaning. For example, 'Laboratory and field experiments on soil freezing/thawing have received more attention' (L54); what does this mean? More than what? Why is this needed? Similarly go to L36: "climate change in cold regions"? What about climate change? How does it relate to solute? Explain. This doesn't fit the sentence. There are many examples of such vague statements with little to no information (e.g., L286-287). I don't see this as an English problem, but rather as a contribution that needs to be carefully rewritten in general.

Re: We can understand this statement but we also strongly feel that the way of expressions from the reviewer is demonstrating and strange attitude to paper but using expressions such as "incredibly vague". However, we do admit that many expressions can be improved and make more precise and add references when needed to justify the statements. We will check them and make the expression much clearer for readers. For example, in L54, 'Laboratory and field experiments on soil freezing/thawing have received more and more attention', in L36, 'climate change induced nutrients loss in cold regions'. We will take this seriously and re-write vague sentences.

3. The introduction does not build a convincing story of why this contribution is needed. First of all it is too long (9 paragraphs). As an example of extraneous text, in Lines 37 to 95 the authors list a number of soil freeze-thaw field and modeling studies and where those studies were conducted. However, they really don't emphasize the contribution or key input of over half of these studies. These are described in far more detail in the review paper by Kurylyk and Watanabe (2013), and listing fewer of these and referring to this synthesis would be a more effective use of space. More importantly, the main objective (e.g. 'search for constrain' L133), makes no sense, and thus it is very hard to get excited about the rest of the paper.

[Figure]

Re: Admittedly, the Introduction part is a little longer, because we wanted to give a good review of experiments and modeling work in soil freezing and thawing. However, some information was not properly conveyed here. We will re-write the Introduction part to make it more concise and related to our work. The main objective is reformulated to emphasize the important link between the solute concentrations and the explicit consideration of the dynamic impact on the freezing point depression.

4. The title does not have 'uncertainty' in it. In fact, the title could probably equally apply to about 12 of the other papers cited in this study. My point is that it should be rewritten somehow to reflect the distinct aspects of this study.

Re: In this paper, we mainly focused on the development of CoupModel by considering the influences of solutes on soil freezing point. Uncertainty analysis was used to calibrate the model and explain the model results. We use the title 'Simulations of water, heat, and solute transport in partially frozen soils' because we wanted to emphasize that in this study we modeled the transport of water, heat, as well as solute in partially frozen soils. This is the difference of this model with previous ones, because for previous models, the influences of solute on soil freezing and thawing were not taken into account carefully or neglected. We do understand that the uncertainty approach is an important aspect of the paper but not as a methodological contribution. The uncertainty is a standard tool to demonstrate to what extent the data could be used to justify the suggested new explicit of considering the dynamic simulated solutes impacts on the freezing point.

5. L168, why did the authors record TDR data at daily intervals? This seems like a low resolution given the frequency of the other data. Depending on the depth and spatiotemporal resolution, it can be very hard to calibrate or assess a model using only daily moisture data.

Re: The TDR data was only recorded in daily intervals in one study site, this is one source of uncertainty in the measurements. This was also discussed in the paper.

That is also why we used Monte-Carlo method to calibrate the model. The model also detected the uncertainty in soil moisture modeling results. This could give us proof that in the next step of experimental design, to focus more on soil water content measurements in soil freezing/thawing period with higher spatial and temporal resolutions. However, the daily resolution is not a major problem when we consider sub-surface processes. The TDR has big difficulties to represent the very shallow soil horizons where the high temporal resolution is mostly pronounced.

6. There are issues with the only two equations I looked at very closely on a term by term basis. Equation 1: the terms in this equation do not have consistent units. Therefore something is clearly wrong. I think the second term on the right (vapor diffusion) has incorrect units and the vapor density should be expressed as a vol/vol. Equation 2: this equation is expressed incorrectly for the divergence of convection (second term on right hand side). The temperature has to be inside the derivative. It is changing with space. Also, if you leave it outside the derivative, you have the issue that it totally depends on what temperature scale you are using (Celsius vs. Kelvin) in terms of the magnitude of that term. All other terms are independent of the temperature scale, because it is only the change in temperature that matters (i.e. the other T terms are inside derivatives). Two errors in the two equation carefully considered does not give one confidence in the rest of the paper. I strongly recommend that the authors go through the equation appendix very carefully.

Re: Thanks for pointing out the units for vapor density, which should be m3/m3. However, this does not mean the model has a mistake in numerical calculation. As to second term in heat transport equation, we agree that it should be inside of the derivative. We do invite all interested reader to check all equations. In addition the model is available as public domain to check additional details. Again, we will check all the equations and units to make them correctly illustrated.

7. The figures are poorly done in general. Is Figure 2 taken from the Coup manual, at least in revised form? If so, that should be stated. Figure 3 is unclear. What is energy?

[Figure]

Is this sensible and latent heat? What do the different colors represent? Figure 4 is confusing. Why is the cross-hatching so similar? Are these cumulative? For example, for time period 2, the 0-100 section goes from 0 to -18 (I think), but the next one goes from -18 to about -38. What does this represent? Explain in the caption!

Re: Fig.2 was plotted by the author, according to the model theory in water, heat, and solute transport. It is original. Lines in Fig.3 will be changed in terms of color and line width. In Fig. 3, energy means total heat storage in soil. Different colors indicate different solute conditions. We will address them in detail in figure caption. In Fig.4, the cross-hatching represents cumulative value. It will be explained in the caption.

8. How does this paper differ from Wu et al. (2016) by the same authors. It is an uncertainty study using a similar sort of model it would seem. I'm not saying there are no differences, but it seems, at least surficially, to be similar. Wu, M., P.-E. Jansson, X. Tan, J. Wu, and J. Huang. 2016. Constraining parameter uncertainty in simulations of water and heat dynamics in seasonally frozen soil using limited observed data. Water 8(2):64, doi:10.3390/w8020064

Re: The paper published on <Water> focused on the uncertainties in parameters in calibration of a water and heat transport model. It is the first step of modeling work, which we investigated parameter uncertainties by using a static simple expression of freezing point depression that was not explicit to the dynamic variability of solutes in the soil. While this paper is focusing on the model performance in considering the dynamic variability of solute on the soil freezing processes. The current paper reflects a new theory compared to the previous paper. The similarity for these two papers is the data used are originating from the same experimental field study.

9. Oddly enough this is never cited. Also, the authors have a lazy reference list. For example, Wu et al. has been published in Cold Reg. Sci. Technol. For over a year now, but the authors list it as accepted. The cited Wu et al. 2016 study has no journal information, so it is not easy to look up. Again, those were the only two reference items

that I looked at.

Re: We are sorry for this mistake. Actually, these papers were not published when writing this one, we did not check them when we submitted this paper. We will check the reference carefully to make sure they are correctly cited.

These are all very short general comments, but a more careful, rigorous review is really difficult given the present state of the manuscript. In summary, this is certainly not publishable in its current form, but I think it does have potential for publication in some journal some day given the content.

Re: Thanks for the constructive parts of your comments. We will revise this paper carefully according to the suggestions from the reviewer, and make it much easier to read without misunderstanding because of vague expressions or poor English.

---

## Author Comment (AC2) · 21 Feb 2017

REF 2 Comments General comments The manuscript sent by Wu et al. provides a study of simulating water and solute infiltration in partially frozen agricultural soil in northern China. Two winter periods were simulated. They used 1D subsurface, surface and atmosphere model (CoupModel) that is capable of simulating heat transfer, and soil water distribution in frozen and partially frozen soil. CoupModel is highly tested and quite well recognize model in cold snow dominated regions, and can be the most sophisticated model assesses water infiltration in partially frozen soils in cold regions. The author uses novel technique in calibrating the highly parameterized model. Even though Monte Carlo method is well used and has been (and should be) a standard process in every simulation exercise this paper provides some new information of sensitivity of parameters affecting simulated soil water content and temperature (heat

transfer) in frozen soils, new insight is also given to account solute in the model cali-bration, although it seems that CoupModel cannot provide a very good solute transport solvers (as author also clearly recognizes this) and there can be better tools available for that. This study also reveals that where the pitfalls are if solute transport is taken into account, and show that the results can be improved if solute transport is taken into account. There are some moderate/major issues that the authors need to take into account prior to publication.

Abstract Lines 24-25, the author mentions "novel techniques" but this study does not explore these techniques in depth, this needs to be removed from the abstract because it is not really the scope of the study.

Re: Thanks for the suggestion, we will remove it.

Introduction In recent literature there has been debate of using impedance factor, that CoupModel uses, in hydraulic conductivity function to account the blocking effect of ice (see Painter et al. 2016; Kuryluk and Watanabe 2013). The author could shortly improve introduction and discuss about this matter, because it has been raised up in the recent literature.

Re: We will re-write this part according to the recent literature suggested by the re-viewer.

REF Kurylyk, B.L., and K. Watanabe (2013), The mathematical representation of freez-ing and thawing processes in variably-saturated, non-deformable soils, Adv. Water Re-sour., 60, 160-177 Painter, S. L., E. T. Coon, A. L. Atchley, M. Berndt, R. Garimella, J. D. Moulton, D. Svyatskiy, and C. J. Wilson (2016), Integrated surface/subsurface per-mafrost thermal hydrology: Model formulation and proof-of-concept simulations, Water Resour. Res., 52, 6062-6077, doi: 10.1002/2015WR018427. In lines 54-73, the author brings up a lot of literature about previous studies but do not mention the most impor-tant results of them. Maybe the most relevant studies with respect to this study should only bring up, and the most important results of them and how they are correlated or

not with this study. The author mentions very lightly (see line 71) the most common results: that soil condition and boundary conditions have an effect but this is way too general outcome of these previous studies. Line 93, AIS model should be mentioned in this list (see Painter et al. 2016, WRR paper) because they have freezing/thawing included in distributed hydrological modeling.

Re: We will read the references carefully and summarize the previous studies focusing on the findings instead of some general outcomes. And the more recent contribution to this subject will be added in the introduction.

Line 103, delete "the" before neglecting

Re: Will be deleted.

Line 104, delete "the" before neglecting

Re: Will be deleted.

Line 121, "," delete empty space

Re: Will be deleted.

Material and Methods Line 187-191, why the observation interval was not denser and not the same as observations from the meteorological station?

Re: Observations for soil moisture and soil temperature was conducted in different parts of the field, which is several hundred meters away from the meteorological station. These field observations were done manually, so it was not possible to keep the same intervals as in meteorological station, which was automatically.

Line 193, delete "theory", because it is not only CoupModel theory but theory in general (flow etc.), maybe the author could invent a new headline.

Re: Will be deleted.

Line 195, why groundwater level was chosen as a lower boundary condition and not

e.g. bedrock?

Re: In this study, we chose the lower boundary at 6 m depth. This was because groundwater level will not reach there. So water content at this depth could be seen as stable as at bedrock.

Line 196, "CoupModel could be ..." what this "could" means? Maybe "could" should be deleted

Re: Will be deleted.

Line 199 and Line 203, check the meaning of kw

Re: kw is unsaturated hydraulic conductivity, we will make the expression consistent in L199 and L203.

L217, the CoupModel neglects diffusion, maybe the author could discuss about the effect of neglecting diffusion a little bit more in detail, the simulated results are not that good and maybe this is the major issue here.

Re: Thanks for good suggestions. We will add discussions on this diffusion in the model, actually it could be a source of uncertainty in the modeling of solute transport. We have realized this when we developed the model and analyzed the modeling results on solute, and also mentioned it. But it would be more detailed explored in the revision.

Line 222-223, is the lower boundary "groundwater" level or "drainage", I think the choice of groundwater level, or drainage may affect the results. If drainage is used, then the drainage equation should be clearly shown. The author only mentions that this equation appears in the table S2.

Re: The lower boundary is drainage. The drainage equation here used is Hooghoudt equation. We did not list it in the main text. We will put it in the main text in the revision.

Line 232, T0?, please refer to equation
Re: T0 refers to Equation (5), we will add it in the revision.

Line 284, what the author mean that you focus on calibration and not validation? What do you mean by validation?

Re: Here, validation means using data outside of the calibration period (e.g. one year for two sites) to assess the model. Since we focused on the model performance in calibration with limited data, we did not discuss the model performance outside of the simulation period in this paper.

Results and Discussion Line 341 How "temperature gradients" affect the soil water and solute transport, please explain.

Re: Temperature gradient could drive water flow in soil profile, which could also cause dissolved solute transport. This will be explained in revision.

Line350, define NP

Re: NP means number of parameters show tight correlation to variables. Will be explained in main text.

Line 365-371, this is difficult to follow, if the reader is not aware of the equations, maybe the equation should be explained in the text, at least a proper references should be provided that reader immediately knows what equations should be looking at in the appendix.

Re: The reference of each parameter to the equations will be added here in main text to make it much clearer.

Line 377-380, what this means in practice? Be more specific.

Re: It means in choice of parameter ranges for these parameters related to evaporation and radiation processes, we should properly decide the prior distribution of parameters as well as their value ranges. This will be addressed more specifically in the main text.

Lines 381-384, what this means in practice? Be more specific.

Re: This means in calibration of model for Site NE, we need to choose proper representation of snow and frost processes with respect to model structure and parameters values. For Site IM, we need to choose proper equations for representing water transport and surface energy balance, as well as reasonable parameters related to these processes.

Line 388-394, the resolution (time interval) of observations seems to be very important, why not taking data hourly?

Re: Unfortunately, due to difficulties in sampling frozen soil and in recording TDR data in study sites, we have limited measurement intervals. We did all measurements manually, so this limited the observation intervals. We may not be able to change them in this study. But we can learn from the study that in the next step of experimental work, we will increase measurement intervals for soil water content during freezing/thawing season.

L400-403, can this be also an issue of the model, see lines 205-207?

Re: This could be both an issue of the model, because in the model some processes are not taken into account, e.g. surface ice cover in Site IM, lateral drainage in Site NE. These could cause uncertainty in modeling results.

Line 436-437, unclear, please reword

Re: Reword as 'Total water content at 5 cm depth shows relatively large uncertainties in both sites. For the whole simulation period, larger uncertainties were also detected when soil started freezing or was nearly totally thawing.'

Line 440-441, unclear, please reword

Re: Reword as 'These parameters will not be suitable for unfrozen conditions, when large water fluxes occur in soil.'

Line 458, "Simulated liquid water content…"? delete "in comparison with the measured"

Re: Will be deleted.

Line 466-467, please reword. The equations should be clearly shown or referred.

Re: Related equations will be referred here in revision.

Line 482, "validation", calibration instead?

Re: Will use 'calibration'.

Line 485 delete "work"

Re: Will be deleted.

Line 485 "surface water"? do you mean "soil water"?

Re: Has changed into 'soil water'

Line 484-487 unclear, please reword

Re: Reword as 'Since the study focusing on water and energy balance in upper 40 cm soil layer, we did not take water and heat transport in lower layers.'

Line 494, delete "work" after calibration

Re: Will be deleted.

Line 496, change "assumption of" to "assumption that"

Re: Will change as suggested.

Line 504-505 unclear, please reword

Re: Reword as 'As is known that, during soil freezing period, water and solute fluxes are mainly upward.'

[Figure]

Line 517 carefully., delete ”.”

Re: Will be deleted.

Line 517-518, be specific that what type of data should be collected

Re: Reword as '. . . e.g. solute concentration, continuous liquid water content measurements would be of importance in calibration of model, . . .'

Line 555 "surface water"? do you mean "soil water"

Re: Changed into 'soil water'

―――――――――――――――――